# Quantifying Circadian Aspects of Mobility-Related Behavior in Older Adults by Body-Worn Sensors—An “Active Period Analysis”

**DOI:** 10.3390/s21062121

**Published:** 2021-03-18

**Authors:** Tim Fleiner, Rieke Trumpf, Anna Hollinger, Peter Haussermann, Wiebren Zijlstra

**Affiliations:** 1Institute of Movement and Sport Gerontology, German Sport University Cologne, 50933 Cologne, Germany; t.fleiner@dshs-koeln.de (T.F.); rieke.trumpf@lvr.de (R.T.); A.Hollinger@dshs-koeln.de (A.H.); 2Department of Geriatric Psychiatry & Psychotherapy, LVR Hospital Cologne, 51109 Cologne, Germany; peter.haeussermann@lvr.de

**Keywords:** circadian motor behavior, body-worn sensors, older adults

## Abstract

Disruptions of circadian motor behavior cause a significant burden for older adults as well as their caregivers and often lead to institutionalization. This cross-sectional study investigates the association between mobility-related behavior and subjectively rated circadian chronotypes in healthy older adults. The physical activity of 81 community-dwelling older adults was measured over seven consecutive days and nights using lower-back-worn hybrid motion sensors (MM+) and wrist-worn actigraphs (MW8). A 30-min and 120-min active period for the highest number of steps (MM+) and activity counts (MW8) was derived for each day, respectively. Subjective chronotypes were classified by the Morningness-Eveningness Questionnaire into 40 (50%) morning types, 35 (43%) intermediate and six (7%) evening types. Analysis revealed significantly earlier starts for the 30-min active period (steps) in the morning types compared to the intermediate types (*p* ≤ 0.01) and the evening types (*p* ≤ 0.01). The 120-min active period (steps) showed significantly earlier starts in the morning types compared to the intermediate types (*p* ≤ 0.01) and the evening types (*p* = 0.02). The starting times of active periods determined from wrist-activity counts (MW8) did not reveal differences between the three chronotypes (*p* = 0.36 for the 30-min and *p* = 0.12 for the 120-min active period). The timing of mobility-related activity, i.e., periods with the highest number of steps measured by hybrid motion sensors, is associated to subjectively rated chronotypes in healthy older adults. The analysis of individual active periods may provide an innovative approach for early detecting and individually tailoring the treatment of circadian disruptions in aging and geriatric healthcare.

## 1. Introduction

Morning lark or night owl—what is your preferred time of the day? The growing knowledge of and interest in the impact of circadian rhythms in daily life refers to circadian medicine [1], where individual chronotypes and circadian characteristics play a key role in society and health care [2].

Physiological processes and behaviors synchronized to a 24 h structure are defined as circadian (lat. circa = approximately, -dian = day) [3,4]. The stability of circadian behaviors is especially relevant in older adults and geriatric health care, where aspects of circadian behavior may show deviations ranging from age-associated changes in subjective chronotypes [5] to clinical syndromes [6]. Disease-related changes of the circadian system occur, for example, as sleep disturbances with reversed day-night rhythms [7], or sundowning phenomena with increased levels of physical activity (PA) and behavioral disturbances in the afternoon and evening hours [8,9]. Disturbed circadian rhythms cause a significant burden for both the patients themselves as well as their caregivers [10] and often lead to institutionalization, especially in home-dwelling dementia care [11].

Within chronobiological research and geriatric sleep medicine, subjective or proxy-based psychopathometric instruments [12,13] and objective approaches like polysomnography or body-worn motion sensors are usually applied to assess circadian characteristics [14]. Most commonly, uni- and multi-axial accelerometers attached to the non-dominant wrist, so called actigraphs [15], are used as ambulatory assessment to quantify circadian motor behavior. The accumulated raw activity counts of wrist movements are analyzed by non-parametric methods—e.g., by deriving an acrophase that refers to the timing of the peak activity within an day [16], or analyzing the intradaily variability, and the interdaily stability of the counts per minute [17]. As these actigraphs only record wrist movements, these measurements and analyses provide a general assumption of temporal aspects of PA and do not enable to detect specific motor behavior patterns. Studies in geriatric care and investigations in community-dwelling older adults indicate the wrist activity to be independent of the distribution of the step count [18,19]. Therefore, such actigraphic measurements do not allow to derive personalized interventions, e.g., like physical activity programs scheduled as circadian zeitgebers [20,21].

Hybrid motion sensors attached to the lower back can detect the patients’ body postures over several days, allowing to analyze individual mobility patterns concerning mobility-related behavior [22]. First studies conducted with older adults have investigated inter-daily walking duration and step count with sensors attached to the participants’ trunk or thigh [23,24]. Up to now, only a few approaches have been developed and applied to investigate the temporal distribution 65 of mobility-related PA in older adults. For example, the investigation led by Lim [18] analyzed the gait activity during the day using the number of active minutes (≥4 steps per minute), and the study reported by Paraschiv-Ionescu [25] analyzed the complexity of motor behavior by barcoding the participants’ motor behavior during the day. Both studies used sensor-based approaches to monitor mobility-related physical activity but did not address chronotypes and circadian aspects of motor behavior. As these mobility-related measurements promise an added value over wrist-worn actigraphs for use in diagnostics and treatment, the primary aim of this study is to investigate the association between the timing of mobility-related active periods and subjectively rated chronotypes in healthy older adults.

## 2. Materials and Methods

### 2.1. Study Design

This investigation was part of the ChronoSense project—a cross-sectional study to investigate the use of body-worn motion sensors to quantify chronotypes in older adults (DRKS00015069, German clinical trial register). The study protocol was approved by the Ethics Committee of the Medical Association North Rhine (registration number 2018192) and the Ethics Committee of the German Sport University Cologne (registration number 156/2017).

### 2.2. Participants

Participants were recruited by sending out emails with information brochures to local senior citizens’ networks and to employees of a large municipal association in the Rhineland region in Germany, and through word-of-mouth referrals. Furthermore, invitation letters were sent to persons who expressed interest in participating in studies of the Institute of Movement and Sport Gerontology in the past. These persons had not participated in any studies in the previous year.

Inclusion criteria for participation in the project were as follows: age of 65 years or older, community-dwelling, a score on the Mini-Mental Status Examination (MMSE) ≥24 [26,27], subjective health (self-reported), no full-time employment and written informed consent regarding the study procedures. Any acute or severe mobility impairment, cardiovascular disorder, cognitive disorder or neurological disease (assessed with the Functional Comorbidity Index (FCMI)) [28], which can interfere with functional mobility, led to exclusion from the project.

### 2.3. Instruments

The self-estimation of chronotype was assessed using the Morning-Eveningness Questionnaire (MEQ) [29]. The MEQ is a self-administered questionnaire, determining the circadian chronotype based on 19 questions concerning the participant’s usual daytime preferences. Five chronotypes are distinguished based on the total score of the MEQ: definite evening type (16–30 points), moderate evening type (31–41 points), intermediate type (42–58 points), moderate morning type (59–69 points) and definite morning type (70–86 points). For further analysis, the moderate and definite evening type as well as the moderate and definite morning type were each grouped together. In order to determine the waking time during the day, the participants were asked to log their get up and got to bed times in a sleep diary [30].

The wrist-worn MotionWatch 8 (Camntech, Cambridge, UK) was used for the actigraphy-assessments. It integrates a triaxial accelerometer, a light sensor and an event marker button. The MotionWatch 8 (MW8) was attached to the wrist of the participants’ non-dominant hand. The participants were asked to push the event marker button when getting up and going to bed. The sample frequency of the accelerometer was 50 Hz. The raw acceleration measurements were processed by the on-board software of the MW8 to produce a quantitative measure of the activity during each epoch. For this, the X, Y and Z-axes of the accelerometer were sampled with filtering at 3–11 Hz. The peak X^2^ + Y^2^ + Z^2^ value was tracked. At the end of each second, the square root of the peak value from that second was calculated. This was compared to a threshold of 0.1 g. Values below this threshold were ignored to simplify the final activity graph. Activity that caused the acceleration signal to exceed the threshold was counted as activity. At the end of each epoch of 60 s, the number of activity counts were accumulated. This value was recorded as the ‘Tri-Axial count’ for the epoch.

The mobility-related measurements were conducted using the Dynaport Move Monitor + (MM+; McRoberts, The Hague, NL). The MM+ consists of a triaxial accelerometer, a triaxial gyroscope (sample frequency for both sensors: 100 Hz), a triaxial magnometer, a barometer and a temperature sensor. Data can be collected for up to seven consecutive days. In order to enable a consistent recording of PA, waterproof self-adhesive foil (Opsite Flexifix, Smith and Nephew, London, UK) was used to attach the MM+ to the participants’ lower back, approximately 3 cm right to the fifth vertebra of the lumbar spine (L5). The participants were asked not to remove either sensor during the measurement period. To ensure an assessment of habitual awake and rest phases, only sensor data of participants with four or more complete measurement days were included.

### 2.4. Data Collection

Data collection covered the period of one week. During an individual appointment in the laboratory, the MEQ was administered and participants were equipped with the two sensors and received the sleep diary. Furthermore, the participants’ living situation (e.g., marital status, income) as well as their health status (e.g., number and kind of chronic diseases) were assessed using a custom-made questionnaire.

At the end of the measurement period, the sensors were removed from the participants’ body. The participants were asked to specify whether or not they had removed one or both sensors during the measurement and to indicate the period if this was the case. In order to ensure that the measurement period represented a habitual week in terms of the participant’s PA and wake and rest periods, special events (e.g., acute illness) were noted.

### 2.5. Data Processing and Statistical Analysis

The MW8 raw data were processed using the validated proprietary MotionWare software (CamNtech, Fenstanton, UK). Total counts per 60 s epochs as well as the getting up and bedtimes based on the event markers set by the participants were included in the output. Average counts per minute were calculated for 24 h. The duration of wakefulness (time from getting up to bedtime) for each day was calculated. In case a participant forgot to set the marker, the corresponding time from the sleep diary was used instead.

The MM+ raw data were processed using the validated manufacturer’s own algorithm (MoveMonitor, McRoberts, The Hague, NL). PA category (walking, stair walking, cycling, shuffling, standing, sitting and lying) as well as the categories not-worn, activity duration and number of steps per 60 s epoch were provided within the output. For the description of this study sample, the average durations of PA types and total number of steps were calculated for 24 h.

In order to quantify circadian aspects of mobility-related behavior, we determined an active period for each day. The active period was defined as the time interval in which the highest PA was measured during wakefulness (from getting up to bedtime). For the MW8, the active period was determined based on activity counts, and the active period of the MM+ was determined based on the number of steps. According to the recommendations of the American College of Sports Medicine to be active for a minimum of 30 min per day on five days per week [31], we chose to determine the active period for an interval of 30 min. Referring to the MEQ, rating the best time of the day for performing two hours of physically hard work, we chose to determine a 120 min active period [29]. Matlab R2020a (The Mathworks, Natick, MA, USA) was used to detect the time of the beginning of each active period. To this purpose, the total number of steps or counts over a time interval of 30 or 120 min was calculated repeatedly, starting with the first available data (when participants got up) and repeated by shifting the start of the time intervals to each next minute. This was repeated until the very last interval (30 or 120 min before the participant went to bed). Subsequently, all intervals were sorted in ascending order and the interval with the highest value (number of steps or number of counts) was determined as the active period. As the sensor measurements were started at 8 pm on day one and ended at 8 pm on day 8, the active periods were analyzed for day two (getting up) to day seven (going to bed). Finally, the mean start times of the 30-min and 120-min active phases were determined for each participant.

IBM SPSS Statistics 26.0 for Windows (International Business Machines, Armonk, NY, USA) was used for statistical analysis. Boxplots were used to identify extreme values. Values of more than three times the interquartile distance were excluded from further analysis. Subsequently, the Kolmogorov Smirnov test was used to examine data for normal distribution. One-way analyses of variance (ANOVAs) or Kruskal–Wallis tests were performed to assess differences in the start times of the active period between the three groups. Bonferroni post-hoc tests were used to examine significant differences. An alpha < 0.05 was considered to be statistically significant.

## 3. Results

### 3.1. Participants

A total of 118 persons were screened with regard to the inclusion criteria. Twenty-three persons declined to participate; 10 persons did not fit to the inclusion criteria. Eighty-five persons agreed to participate. Two participants became ill during the measurement period and were excluded from data analysis. One participant indicated that he was less active than usual during the measurement period, and one participant did not wear the sensors according to the instructions. The data of these two participants were excluded from analysis. Finally, the data of 81 participants were analyzed. Table 1 shows their characteristics.

MM+ data were available for 75 (93%) participants. The MM+ data of six participants (7%) were missing due to technical problems. Six complete measurement days were available for 67 participants (83%). Seven participants (9%) completed five measurement days. One participant (1%) completed the minimum requirement of four measurement days. MW8 data were available for 66 (82%) participants. MW8 data of 15 (18%) participants were missing due to technical problems. Six complete measurement days were available for all 66 participants.

The distribution of self-estimated chronotypes and the corresponding characteristics of subgroups is shown in Table 2. Statistical analysis revealed no significant differences between groups in the sample characteristics and their general level of PA.

### 3.2. Active Period Analysis

Figure 1 shows the comparison of the summed 30 min time intervals of the number of steps per minute for one subject from each chronotype group. The time of day at the peak of each curve indicates the beginning of the 30 min active period based on the step count for one participant of each chronotype group.

Significant differences within the circadian aspects of the step count (MM+) between the three groups were detected (Figure 2 and Figure 3).

Compared to the morning type group, the intermediate type group showed a delay in their 30-min active period of approximately one hour (*p* ≤ 0.01) and the evening type group a delay of approximately two hours (*p* ≤ 0.01), respectively. These results were also found for the 120-min active period. Compared to the morning type group, the intermediate type group showed a delay of approximately one hour (*p* ≤ 0.01) and the evening type group a delay of approximately two hours (*p* = 0.02), respectively. No differences within the circadian aspects of the activity counts (MW8) were detected regarding the three chronotype groups for both the 30- and 120-min active periods.

## 4. Discussion

The primary aim of this study was to investigate the association between the timing of mobility-related active periods and subjectively rated chronotypes in healthy older adults. The analysis revealed significant differences in the starting times for the 30-min and 120-min active period (steps) between the chronotypes. The starting times of the active periods regarding the wrist-activity counts did not reveal differences between the three chronotype groups.

The “active-period” analysis is a novel approach in this field of research. Whereas the usually applied wrist-worn actigraph approach showed no differences in activity-related behaviors over the three chronotype groups, this study’s results reveal the hybrid motion sensor to be able to quantify circadian aspects of mobility-related behavior, i.e., regarding the number of steps. The timing of the peak wrist activity within a day, usually reported as acrophase for wrist-worn actigraphy [16], seems to be independent of the timing of the peak gait activity, reported as active period. These differences between objectively measured wrist-based activity (counts) and mobility-related behavior (steps, postures) are comparable to previous results [19]. Additionally, studies applied in an acute geriatric care setting reported three peaks in the wrist-measured physical activity at 9 am, 12 pm and 5 pm, referring to the patients’ meal times and showing no relation to the distribution of the step count within the patients’ day [18]. As compared to the usually used wrist-worn actigraphy approach in chronobiological research and geriatric sleep medicine, analyzing the active period of mobility-related behavior seems to provide more clinically relevant and essential information. Such circadian aspects of mobility-related behavior could be applied to assess circadian disruptions based on the temporal distribution of the step count within a day and subsequently derive, individually tailor and evaluate interventions to treat circadian disruptions, e.g., exercise approaches based on step counts [20,21,32].

The participants included in this study were healthy, community-dwelling older adults, on average 72 years old (SD 5), with a daily step count ranging from 3278 to 17,320 with on average 9860 steps per day (SD 3280). These activity measures reveal a general active lifestyle, as the endorsed level of 7000–10,000 steps per day was almost achieved in this group [33]. With a mean of 317 activity counts per minute [24 h] (SD 87), the study sample shows comparable levels to other studies using the same actigraphy approach [34]. The included participants subjectively rated themselves mainly as morning type (*n* = 40, 49.4%) and intermediate type (*n* = 35, 43.2%), but only six participants rated themselves as evening type (7.4%). This distribution of chronotypes is comparable to previous studies, which reported more morning types in association with higher age [5].

An analysis and interpretation of this study and its results should consider the following methodological limitations: established by Horne and Ostberg [29], the Morningness-Eveningness Questionnaire usually categorizes five chronotypes. The definite and moderate morning- and evening types were accumulated in order to analyze differences between the three main chronotypes. The current analysis did not reflect inter-daily consistency of active periods. Future analysis should address these aspects, e.g., via coefficient of variance. A potential selection bias should be taken into account, as the sample has been shown to be highly active with approximately 10,000 steps per 24 h.

The results of this study contribute to the growing knowledge and interest on the impact of circadian rhythms in daily life and healthcare [1,2]. Analyzing the starting times of the active periods for mobility-related behavior, e.g., by the number of steps measured by hybrid motion sensors (MM+), seems to be a clinically relevant approach to quantify circadian aspects in healthy older adults. Analyzing circadian aspects of mobility-related activity, and potentially also temporal patterns of inactivity, could play a key role in aging research and geriatric healthcare, especially in the assessment and treatment of circadian disruptions. In addition to the presented results of not showing differences in the assessment of active periods, the wrist-worn actigraphy approach (here MW8) does not allow to derive, apply and evaluate individually tailored interventions. This is essential for its clinical application, and therefore limits its use in general and especially in geriatric healthcare [19]. The presented “active period analysis” provides an innovative and clinically relevant approach to quantify circadian aspects of mobility-related behavior with body-worn sensors in older adults. Especially in patients suffering from circadian disruptions, an individual (in)active period could be used to derive, apply and evaluate step-based interventions [35] potentially combined with day-structuring approaches. The individual active period analysis may improve the early detection and individual tailoring in the treatment of circadian disruptions in aging and geriatric healthcare that may have promising effects for patients, caregivers and geriatric healthcare [1,2].

## Figures and Tables

**Figure 1 sensors-21-02121-f001:**
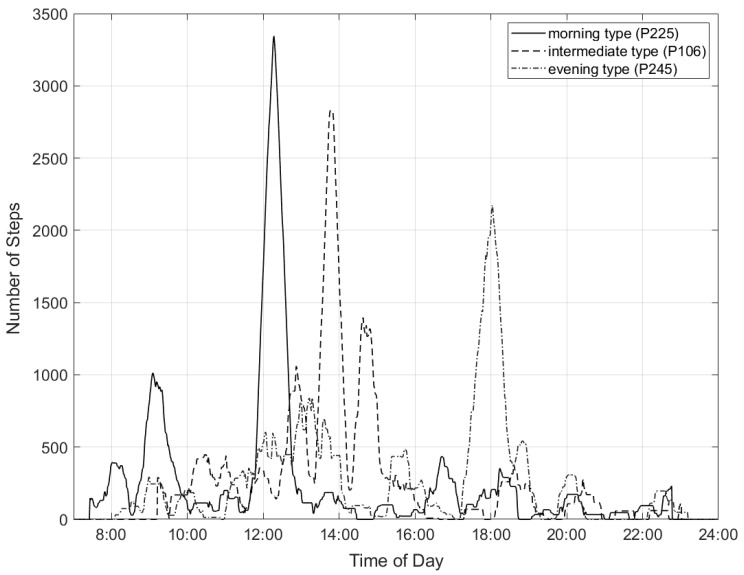
Exemplary analysis of 30-min active period.

**Figure 2 sensors-21-02121-f002:**
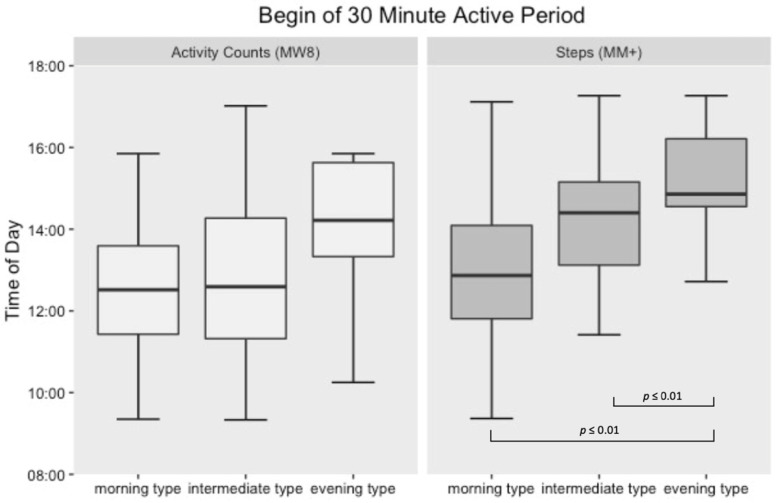
Box-plot illustration of 30-min active periods beginnings in relation to the subjectively rated chronotypes.

**Figure 3 sensors-21-02121-f003:**
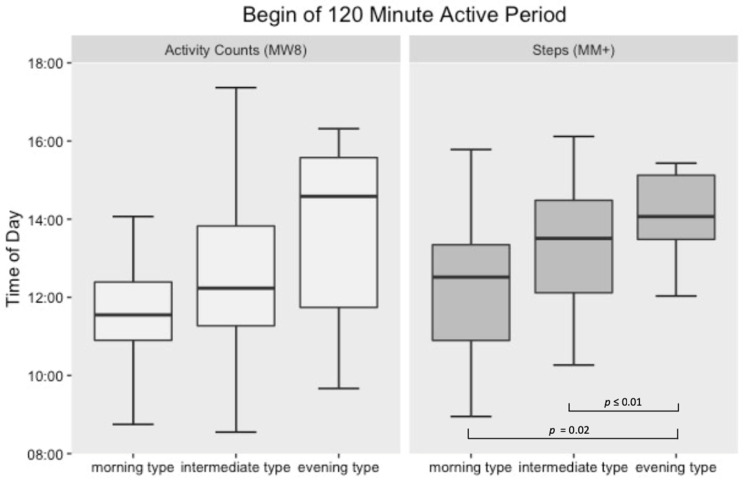
Box-plot illustration of 120-min active periods beginnings in relation to the subjectively rated chronotypes.

**Table 1 sensors-21-02121-t001:** Sample characteristics.

			N (%)	Mean	SD	Min	Max
Sample			81				
	Female		40 (49.4)				
Age (years)				71.5	5.0	65	84
Mass (kg)				76.9	15.4	54	119
Height (cm)				170.3	8.3	154	188
MEQ score				57.7	9.9	31	77
Number of diseases				2.0	1.4	0	7
Duration of wakefulness (h)				15.9	0.8	13.5	18.1
Move monitor+			75				
	Activity/posture [hh:mm]/24 h						
		lying		09:01	01:38	06:18	14:29
		sitting		09:13	01:53	05:08	14:13
		standing		03:02	00:48	01:02	04:48
		shuffling		00:29	00:10	00:11	01:11
		walking		01:57	00:36	00:44	03:51
		other activities *		00:05	00:11	00:00	00:59
		not worn		00:13	00:25	00:00	03:07
	steps/24 h			9860.1	3279.9	3278.0	17,319.2
MotionWatch 8			66				
	counts/min [24 h]			317.9	87.3	121.6	556.9

MEQ—Morningness-Eveningness Questionnaire (assessment of subjective chronotypes; scores can range from 16–86. Scores of 41 and below indicate “evening types”. Scores of 59 and above indicate “morning types”. Scores between 42 and 58 indicate “intermediate types”); * summation of total activity durations for cycling and stair walking.

**Table 2 sensors-21-02121-t002:** Sample characteristics of self-estimated chronotype subgroups.

		Morning Type			Intermediate Type			Evening Type			*p*
		N (%)	M	SD	N (%)	M	SD	N (%)	M	SD	
Sample		40 (49.4)			35 (43.2)			6 (7.4)			
	female	15 (37.5)			22 (62.9)			3 (50.0)			
Age (years)			72.1	5.2		71.4	5.1		67.8	3.1	0.10
Wakefulness (h/day)			16.1	0.6		15.8	0.8		15.9	0.8	0.55
MM+ Steps (number/24 h)		37	9791.2	3157.1	32	9823.9	3101.1	6	10,478.4	4310.3	0.98
MW8 Counts/min (24 h)		30	316.8	73.9	30	319.1	103.2	6	317.9	40.0	0.98

MM+ Move Monitor+, MW8 MotionWatch 8.

## Data Availability

The data presented in this study are available on request from the corresponding author, upon reasonable request. The data are not publicly available due to privacy/ethical restrictions.

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
