# Peer review of "Quantifying Circadian Aspects of Mobility-Related Behavior in Older Adults by Body-Worn Sensors—An “Active Period Analysis”"

_sensors, 2021, doi:10.3390/s21062121_

Round 1

Reviewer 1 Report

This work investigates the association between mobility-related behavior and subjectively rated circadian chronotypes in healthy older adults by using back-worn sensors and wirst-worn sensors. However, the research seems lack of depth. The reviewer would like to resubmit their manuscript to other journals. The questions to the authors are listed below:
1. Why there are differences between wrist-worn sensors and back-worn sensors in determining starting times of the active periods? Is this because of no gyroscope in wrist-worn sensors? What if participants were measured by using a wrist-worn sensor with accelerometer and gyroscope and advanced algithmn, such as an Apple Watch?
2. What is the reason of differences in starting times of the active peroids between three chronotype groups? The mean values 12:54, 14:13, 15:08 of starting times of the active peroids for three chronotype groups seem to indicate that their most active periods happens after lunch, so are the values associated with their lunch times?
3. Original data. The author should list all experiment data in a graph rather than only a mean value so that the reviewers can know the data distribution.

Reviewer 2 Report

In section 2.3, you describe that the MW8 sensor has a sample frequency of 11Hz collected in 60 second epochs. And in section 2.5 you describe how raw data from the MW8 sensor was processed from 60 second epochs. Did you use a sample rate of 1Hz - or 1 sample per 60 seconds for all activity and sleep measurement from the MW8 sensor? Please clarify more clearly whether 1 sample per minute was captured from the sensor during all recordings of both activity and sleep.

You did not state the sample rate you used for the MM+ sensor. Did you set the sample rate to 1Hz, or did you downsample the data that was output from the sensor? Please clarify your logic around how you decided on the correct sample rate for this sensor so that it was in agreement with the MW8 sensor. If both sensors used differing sample rates, surely this meant that the MW8 sensor was always going to have less data granularity and therefore was going to show less accurate PA levels? Please clarify.

An issue you need to clarify is whether the differences in sample rate had an effect on the accuracy of activity detection. For example, if a person wearing the MW8 sensor walked for 20 seconds and then sat for the remaining 40 seconds, how did that affect the physical activity? How did you deal with this issue? Please clarify by adding a section to describe whether this was an issue, and how you controlled it.

The data shown in Table 3 is unclear. Please explain more clearly what “counts” refers to for MW8. Are you referring to physical activity? And does the time shown represent the start time for each subsection fo activity? If so, please discuss whether the variance in sample rate between both sensors had any effect on accuracy of detection of start time.

Several journals report that slow activity such as shuffling is difficult to detect with wrist worn sensors such as IMU’s. Please comment in the Discussion section your opinions on this and whether it had any effect on the segmentation of activity and posture in your study.

Minor pint – I assume the word “material” on line 122 should be “marital”. Please change.

Reviewer 3 Report

This is a well-written and clearly understandable manuscript in the field of circadian aspects of physical activity / motor behavior during older adults’ daytime. All formal necessities are fulfilled, there are no major scientific flaws (at least none I can identify). Although not a ‘native’ in this field of research, my opinion is that this work’s contribution to the field is significant, and that the question at hand is worth investigating. However, I do see some points which need be addressed to allow this to be published.

1) I see that active period cut-off of 30 min was chosen based on the Chodzko-Zajko paper. As – I am sure – the authors know, there have been updates on physical activity recommendations. The threshold of 30 minutes per day is not being promoted anymore (see https://bjsm.bmj.com/content/54/24/1451#T4). The authors state in their discussion that “analyzing starting times of the active periods for mobility-related behavior […] seems to be a clinical (needs correction: clinically) relevant approach…”. If several shorter intervals than 30 minutes would be used (e.g., 10 minutes), this would allow capturing activities with more granularity over the whole day, and not only a single interval would be taken into account. And it would be in line with current recommendations (see above) stating that basically any physical activity is important, irrespective of its duration. My problem with that is that some persons may be morning larks, but may not have their many activity during this time of day and only become active later, for whatever reason (exercise group, appointments, …). In this way, there would be some sort of misclassification. Why not look at all the active periods (could be defined as more than x steps per minute or x number of consecutive steps, or else) and see whether there are differences between groups here? Only looking at one interval to me seems a “waste” of a lot of data and information. If I understood correct, intervals shorter than 30 minutes will not be taken into account with the current approach?!

2) regarding figure 1, I think it would be helpful to plot averages of all 3 groups, instead of showing 1 subject per group. I have to say, though, that I cannot clearly identify the ‘course of the start times of the 30 min active period’ (line 187). This should be better explained in the text. And why not only plot this very period then? For the morning type, there are actually 2 peaks – is only the highest peak of relevance? This would also cross my first comment. All in all, I think some more explanation of what is shown here is needed.

3) I am having trouble with the fact that a group of only 6 persons is included in the analysis. I would recommend to see this rather a proof-of-principle type of study, which could show the differences between clearly distinguishable groups (morning vs. intermediate), and the third group would not necessarily be required. In this way, a cleaner (and clearer) picture of the differences between groups could be drawn. In short: I would get rid of the evening type group. It confuses the data more than it helps understanding what is going one here.

4) first paragraph (lines 31-34): Would not use direct speech.

5) why exactly was the 120-minute active period used? In which way does the MEQ “recommend” (or whatever you mean here) that? I would like to see a stronger rationale for this choice.

6) I would expect information on how many participants actually wore the sensor over the full measurement period (average wear time, no. of protocol completers, no. of persons with the minimum requirement of 4 days).

7) in line 119 it is stated that data collection covered 1 week; in the abstract, this period is six days. Which one is it?

8) line 220 – 226: this one sentence covers 7 lines. Please rephrase and cut into shorter sentences.

9) My last comment is that the study sample is not really extreme in terms of circadian disruptions, but seems rather normal. This makes me wonder whether interventions to treat circadian disruptions would be needed in this sample? I guess not?! If so, then, the present data would not be applicable to samples with actually disruptive rhythms who would be in need of intervention. What I am referring to is line 266f (last sentence of this manuscript). Detecting clinically useful group differences in such a ‘rather normal’ sample would not necessarily mean that this would be the case in extreme samples actually being in need of intervention. I would tempt the authors to discuss this or argue why they think this is still appropriate.

Reviewer 4 Report

The objective of the proposed study was to investigate the association between the timing of mobility-related active periods and subjectively rated chronotypes in healthy older adults. For this purpose physical activity of 81 older adults was measured over six consecutive days and nights using lower-back-worn hybrid motion sensors and wrist worn actigraphs. Next, a 30-min and 120-min active period for the highest number of steps and activity counts was derived for each day, respectively. The authors through statistical analyzes came to the conclusion that there are significant differences in the starting times for the 30-min and 120-min active period (steps) between the chronotypes. The starting times of the active periods regarding the wrist-activity counts did not reveal differences between the three chronotype groups considered.

The paper  is sufficiently well written, the reading is simple and smooth. It should be emphasized, however, that the contribution of innovation that this work provides to the scientific community is not well highlighted.

A limit is related to the description of the state of the art, which is entirely enclosed in the following sentence: "Up to now, only few studies have investigated the temporal distribution 65 of mobility-related PA in older adults [18,25]". In my opinion, strengths and weaknesses should be described for each paper mentioned.

Moreover, in "Discussion" section the authors state that "the “active-period” analysis is a novel approach in this field of research". Consequently, in section "Materials and Methods"  it would be appropriate to introduce a specific section to describe in more detail the novelties introduced with respect to the scientific literature by this approach.

Some minore issues:

It is necessary to introduce references (possibly also links to websites) for:

1) The wrist-worn MotionWatch 8 

2) Dynaport Move Monitor + MM+ 

3) Waterproof self-adhesive foil 

4) MotionWare software

5) IBM SPSS Statistics 26.0 for Windows 

Round 2

Reviewer 1 Report

The manuscript has been improved,  and I recommend to accept this manuscript.

Reviewer 3 Report

Sufficient for publication.

Reviewer 4 Report

Following the reviews of 4 reviewers I think that the manuscript
has been improved. I recommend to accept
the manuscript in the present form.